# Factors Associated with Undertaking Health-Promoting Activities by Older Women at High Risk of Metabolic Syndrome

**DOI:** 10.3390/ijerph192315957

**Published:** 2022-11-30

**Authors:** Jagoda Rusowicz, Anna Serweta, Karolina Juszko, Wojciech Idzikowski, Robert Gajda, Joanna Szczepańska-Gieracha

**Affiliations:** 1Department of Physiotherapy, Wroclaw University of Health and Sport Sciences, 51-612 Wroclaw, Poland; 2Department of Physical Education and Sport Sciences, Wroclaw University of Health and Sport Sciences, 51-612 Wroclaw, Poland; 3Gajda-Med District Hospital, 06-100 Pultusk, Poland; 4Department of Kinesiology and Health Prevention, Jan Dlugosz University, 42-200 Czestochowa, Poland

**Keywords:** physical activity, metabolic syndrome, health-promoting education, depressive symptoms, obesity, public health

## Abstract

Background: The complexity of health problems concerning women aged ≥60 years makes it necessary to develop effective, low-cost strategies involving biopsychosocial interventions. The aim of this study is to identify the factors associated with undertaking health-promoting activities by older women at high risk of metabolic syndrome (MetS) with or without depressive symptoms. Methods: The study group consisted of 70 older women (62–84 years old) undertaking regular physical activity. A self-developed questionnaire (used to determine the living situation, selected lifestyle components and health problems), the Perceived Stress Questionnaire (PSQ) and the Geriatric Depression Scale (GDS) were used. Results: In the study group undertaking regular physical activity, 40% had increased symptoms of depression (D group), and 60% were classified as non-depressed (ND group). The D group had a higher general stress level (t = −6.18, *p* = 0.001). Improving and/or maintaining physical fitness was identified as the greatest motivation in both groups. Willingness to spend time with other people significantly differed between the two groups (χ^2^ = 4.148, *p* = 0.042). The sole factor significantly differentiating between both groups was lack of time (χ^2^ = 8.777, *p* = 0.003). Conclusions: Motivations and barriers to undertaking health-promoting activities and levels of perceived stress were significantly different between the depressed and non-depressed groups. It is important to encourage primary care physicians to perform screening tests for late-life depression and to provide information on where therapeutic interventions are available for patients with symptoms of MetS and coexisting depressive symptoms.

## 1. Introduction

The world is undergoing unprecedented changes due to declining fertility and mortality rates in most countries and due to the ageing of populations [1,2]. Non-communicable diseases, which are the greatest burden on global health, are more likely to affect adults, including older adults [3]. Data from the multi-country Global Burden of Disease project and other international epidemiological studies indicate that the health problems associated with wealthy and ageing societies affect a wide and growing segment of the world’s population. In every region of the world, people face a higher risk of death and disability from diseases such as diabetes, heart disease and cancer than from parasitic diseases and infectious [3,4,5].

One of the main risk factors associated with the occurrence of cardiovascular disease (CVD) is age. By 2030, approximately one fifth of the world’s population will be ≥65 years old, which will result in a sharp increase in the prevalence of CVD. Environmental factors (over-nutrition, smoking, pollution and sedentary lifestyle) can lead to the premature disruption of mitochondrial function, insulin signalling, endothelial homeostasis and redox balance, promoting features of early ageing [6]. The prevalence of metabolic syndrome (MetS) and diabetes is significantly increasing in the older adult population, especially in women, which further contributes to cardiovascular morbidity and mortality [6,7,8].

MetS is defined as a coexistence of interrelated metabolic risk factors, such as high blood pressure, visceral obesity and carbohydrate consumption and lipid metabolism abnormalities [9]. MetS increases the risk of CVD, which continues to be the most common cause of death in industrialised countries [10,11,12]. Of the adult European population, 10%–30% have MetS, and this number is predicted to increase [13].

The difficulties in treating MetS are related to its multifactorial nature, in which environmental, genetic and psychosocial factors interact through complex networks [14]. The main modifiable risk factors that contribute to the pathogenesis of MetS and its cardiometabolic consequences are obesity, poor dietary patterns and physical inactivity [15,16,17,18]. Psychological stress is also a factor associated with the occurrence of MetS [19,20]. Structured lifestyle interventions and improved diet are essential to the prevention of disease progression. They are currently the first line of treatment for MetS [15,16,17,21,22,23]. Physical activity interventions have a beneficial impact on metabolic disease and its associated burden not only for individuals but also for health systems [24].

Depression is one of the most common and widespread mental disorders in the world [25]. Its pathogenesis consists of interacting genetic, biological and psychological factors [26]. Depressive disorders in the geriatric age range are found in 3.0%–4.5% of the population [27]. In older people, depressive symptoms may be very heterogeneous and often almost imperceptible, which leads to a high risk of late intervention [28]. Depressive symptoms occur mostly in people with poor health as well as incapacitated and widowed people, and women are much more likely to experience depression than men [29]. Older adults with depression have impaired neuropsychological health, greater dependence when performing activities of daily life and higher morbidity and mortality [27,28,30].

A biopsychosocial approach deserves special attention for the treatment of depression in older people [27]. This approach assumes that illness or the disease outcome is the result of a complex combination of factors of different characters: biological, psychological and social [31]. Therapeutic activities in this approach focus on the group nature of all activities, physical training and both psychoeducation and health promotion. This holistic therapeutic approach should be implemented on a long-term basis by a multidisciplinary team of experienced professionals [32]. The biopsychosocial model can lead to improved clinical outcomes by building awareness of the interactions between biological, psychological, sociocultural and spiritual factors, as well as to an increase in a patient’s self-management of illness through a dynamic patient–primary care physician relationship and a multidisciplinary approach to patient care [33].

This study was carried out as part of the Mental Health Promotion Programme at the Foundation for Senior Citizen Activation SIWY DYM, which focuses on biopsychosocial support for women over 60 years of age. The activities offered include regular physical activity, relaxation, health-promotion education, running support groups and social activation to improve health. A previous study demonstrated that low-intensity physical exercise used in a group setting combined with health-promoting education and psychoeducation resulted in a reduction in depressive symptoms by approximately 37% (*p* < 0.01), and the level of self-reported stress and stress level components decreased by 23% and 20% (*p* < 0.01), respectively, in older women with MetS [30]. A subsequent study on a similar group additionally incorporating dance sessions and encouragement to modify diet resulted in a significant reduction in stress intensity (*p* < 0.01) in women with MetS [31].

Research on MetS focuses on modifiable factors such as well-being, physical activity and dietary habits [19,20,34,35] and risk factors (mostly alcohol consumption, smoking, genetic susceptibility, irregular sleep patterns [23,36,37,38]), but rarely addresses the motivations and obstacles that influence making lifestyle changes in older women. This study is focused on understanding the needs of older women who have attempted lifestyle changes to determine exactly what factors may influence the undertaking of regular health-promoting activities in those who enrolled in the Mental Health Promotion Programme.

The aim of this study is to identify factors associated with taking health-promoting actions by older women with or without depressive symptoms at high risk of MetS.

## 2. Materials and Methods

### 2.1. Study Design

This study was observational. It was carried out in a group of 70 women aged ≥60 years (70.4 ± 5.10 years), who applied to participate in the Mental Health Promotion Programme at the Foundation for Senior Citizen Activation SIWY DYM in Wroclaw, Poland. The therapeutic programme consists of general fitness exercises, health education sessions and psychoeducation, where participants are also encouraged to modify their diet. The project has continuously started a new edition every year since 2016 [32,39,40,41].

All patients received a referral to a treatment programme for high risk of MetS and permission to participate in moderate intensity physical training from their primary care physician. Body composition and anthropometric measurements (body height, weight and waist ratio) as well as blood pressure and laboratory tests (e.g., blood results, fasting sugar levels) were ordered by a primary care physician only once, at the time of project recruitment. Approval to conduct the study was obtained from the Bioethics Committee of the Wroclaw University of Health and Sport Sciences in Wroclaw (No. 160614). Informed consent was obtained from all of the subjects involved in the project. The project received funding from the Municipality of Wroclaw. The measurements and analyses were performed by a psychologist experienced in this area.

### 2.2. Inclusion Criteria

The criteria recommended by the International Diabetes Federation (IDF) were used to assess high MetS risk. The criteria are: central obesity (waist circumference ≥80 cm in females), reduced high-density lipoprotein (HDL) cholesterol (50 mg/dL in females), elevated triglycerides (>150 mg/dL), elevated blood pressure (BP; systolic BP > 130 mmHg or diastolic BP > 85 mm Hg) and increased fasting plasma glucose (>100 mg/dL) [42,43]. The occurrence of two of the five characteristics listed above qualified subjects for the project. The exclusion criteria included disturbed cognitive functions (Mini-Mental State Examination >23) [44], a motor disability precluding exercise or the inability to move independently and serious neurological or orthopaedic conditions.

According to the European Society of Cardiology and the European Society of Hypertension (ESH/ESC European Guidelines 2018), the classification of normal BP and hypertension was based on the limit value of 140/90 mmHg with a division into three groups (optimal <120/80 mmHg; normal up to 129/84 mmHg; high normal up to 139/89 mmHg) and distinction of isolated systolic hypertension (ISH) [45,46]. Patients were encouraged by their primary care physicians to participate in the project based on their medical examination results. We do not have information on the number of people who were screened but did not meet the criteria for inclusion in the project.

### 2.3. Participants

The study group consisted of older women at high risk of MetS living in Poland who, after being informed by their primary care physician that they met the inclusion criteria, familiarised themselves with the Mental Health Promotion Programme at the Foundation for Senior Citizen Activation SIWY DYM and decided to voluntarily participate in the study. The characteristics of the study group are presented in Table 1. A Geriatric Depression Scale (GDS) score of > 9 was associated with depressive symptoms, and the participant was allocated to the D group (depressive symptoms group). The other participants were allocated to the non-depressed group (ND group).

### 2.4. Measurement

The GDS, developed by Yesavage et al. [47] for screening the severity of depression in older adults, was used to assess depressive symptoms and depression. The scale contains 30 short questions with two possible answers (‘Yes’ or ‘No’). A score between 0 and 9 points indicates no depressive symptoms, a score between 10–19 points indicates mild depression, and a score greater than 19 points represents a severe form of depression. The reliability of the GDS was assessed using the Cronbach’s α and the Spearman–Brown split-half reliability formula. Cronbach’s α was 0.94, and an identical coefficient value (r = 0.94) was obtained in the split-half reliability measurement of this instrument. The sensitivity and specificity of the GDS were 84% and 95%, respectively [48,49].

Stress levels were assessed using the Perceived Stress Questionnaire (PSQ) by Plopa [50]. The questionnaire, designed to measure the structure of stressful experiences, allows for calculating an overall score, which indicates the overall level of stress, as well as three scores relating to the following dimensions: emotional tension, external stress and intrapsychic stress. It consists of 27 statements for which the respondent identifies the degree to which a given statement concerns him or her, using a 5-point Likert scale ranging from ‘True’ to ‘False’. Each area includes seven statements; therefore, the respondent can score between 7 and 35 points in each area. Subsequently, all of the points are added together to calculate an overall score, which can range from 21 to 105 points. The remaining six statements in the questionnaire relate to the lying scale (from 6 to 30 points). The raw score is converted into a sten score for gender and age, respectively. The sten scores are interpreted as the degree of intensity of the measured trait in the desirable or undesirable direction from the perspective of psychological theory. A sten score of 7–10 indicates a feeling of increased nervousness, anxiety and problems with relaxation. A sten score of 5–6 indicates average intensity of emotional tension. A sten score of 1–4 indicates no emotional strain. The results of the individual stress components are interpreted in a similar way. The internal consistency rates for the three scales (dimensions) in the examination of adults were between 0.70 and 0.81. The factor validity of the PSQ has been confirmed [50].

Weight and height were measured in all participants to determine the BMI. A Tanita BC-545N analyser was used for body composition analysis. In addition, a self-developed questionnaire was used to determine the living situation, selected lifestyle components and health problems in the study group. The questionnaire consisted of 36 questions. The first section concerned age, education and family life (4 questions). The second section referred to the presence of medical conditions and their treatment and complaints of pain (12 questions). The third section related to the recommendations received from primary care physicians, the presence of concerns regarding the respondents’ own health and their satisfaction with their physical condition and weight and the availability of free rehabilitation services in primary healthcare (9 questions). The last section included questions concerning the reason why the respondents decided to participate in the project (6 questions) and the obstacles that have prevented them from taking care of their health so far (5 questions). The questionnaire was completed by the participants prior to the start of the therapeutic programme. The questionnaire, translated from Polish, can be found in the Appendix A.

### 2.5. Data Analyses

Statistical analysis was performed using the STATISTICA 13.3 software from TIBCO Software Inc. (StatSoft Poland, Krakow, Poland). The threshold for statistical significance was set at *p* < 0.05. The traits with continuous distributions are presented as the mean, standard deviation and the range. In addition, distributions of continuous characteristics are presented as distribution series. The homogeneity of the variances was verified by Levene’s test. The normality of the distribution of continuous characteristics was determined using the Shapiro–Wilk test. The null hypothesis of the normality of distribution was rejected for most of the characteristics in the study group. Therefore, the Mann–Whitney U Test was used to compare quantitative and ordinal variables (education level, HDL level, etc.) between the D group and the ND group. A correlation analysis was also performed using the non-parametric correlation coefficient (ρ) to determine the presence of relationships between characteristics (age, BMI, individual components of MetS, GDS and PSQ). A Student’s *t*-test for independent samples was performed to examine stress levels in the D and ND groups. The chi-square test was used to determine the presence of differences between nominal and dichotomous variables (diabetes and hypertension comorbidity, treatment, smoking, etc.) related to the presence or absence of depressive symptoms. Considering the confidential nature of the data analysed in this study (protected medical information), they are available from the authors upon request.

The sample size for our study was determined a priori as a minimum of 40 subjects, based on a confidence level of 95%, a fraction size estimated at 40% and an assumed maximum error of 15%.

## 3. Results

### 3.1. Sociodemographic and Clinical Data

The following health problems emerged in the study group: hypertension (67.1%), prediabetes and diabetes (41.4%), high cholesterol (68.6%) and high triglycerides and hypertriglyceridaemia (38.6%) (Table 2). Only 8.6% of the respondents declared they smoked. In terms of pain, 84.3% reported back pain, 91.4% reported joint pain and 30% reported other pain.

A majority of participants (66%) had concerns about their own health, and 34% had no concerns. The participants were questioned about receiving advice from their primary care physician regarding lifestyle changes, weight loss and dietary changes. The results are shown in Figure 1. The vast majority (78.6%) of the respondents were dissatisfied with access to rehabilitation services offered by the primary healthcare system (the National Health Fund).

### 3.2. GDS-30 and PSQ

The mean GDS score was 7.9 ± 5.17, with a minimum of 0 and a maximum of 22, indicating a low prevalence of depression and depressive symptoms. Considering the detailed GDS results, 42 participants (60%) scored in the range of 0–9, which corresponds to a lack of depressive symptoms; 27 participants (38.6%) scored in the range of 10–19, which corresponds to a mild level of depression; and 1 (1.4%) scored over 20, indicating a severe level of depression. The overall PSQ raw score was 53.5 ± 14.97, with a minimum of 21 and a maximum of 85. This corresponds to a sten score of 3, which is a low level of perceived stress. The Student’s *t*-test for independent samples was conducted to test whether co-occurrence of depression had a significant effect on the PSQ scores. Women who scored ≥10 on the GDS scale had significantly higher scores on the perceived stress scale (t = −6.18, degrees of freedom (df) = 69, *p* < 0.001) (Figure 2).

### 3.3. Analysis of Groups with and without Depressive Symptoms

Detailed analysis showed no significant differences between the D and ND groups in terms of age, education level and family life form. A summary of the data is presented in Table 3.

The mean BMI was quite similar in both groups, averaging 29.6 kg/m^2^ in the ND group and 29.0 kg/m^2^ in the D group. An interesting aspect is the presence of co-existing conditions and information on whether treatment had been provided in the study group. In the D group, there was a noticeably smaller percentage difference between the presence of an illness and whether treatment had been received. In contrast, those in the ND group were less likely to undertake treatment for pre-existing health problems. However, these differences are not statistically significant. A summary of the data is presented in Figure 3.

Health concerns were higher in the D group (75%) compared with the ND group (60%). The prevalence of back pain, pain in other joints and other pain complaints was slightly higher in the D group (86% vs. 83%; 93% vs. 90% and 36% vs. 26%, respectively). Satisfaction with both physical condition and weight also appeared to be similar in both groups (55% vs. 46% and 31% vs. 29%, respectively). There were higher percentages in participants from the ND group, with 55% reporting satisfaction with their physical condition and 31% reporting satisfaction with their weight. In the D group, the values were 46% and 29%, respectively. When analysing the components of MetS, two risk factors were more frequent in the ND group relative to the D group (43% vs. 36%). In turn, a higher percentage of subjects in the ND group had three or more factors (57% vs. 64%).

Participants in the D group relative to those in the ND group were less likely to be encouraged by their primary care physician to reduce their weight (52% vs. 46%). Slightly differently, 61% of the D group and 50% of the ND group were advised to change their diet. Both groups were equally often encouraged to make lifestyle changes (50% each). Spearman’s rank order correlation analysis showed a strong association only between the GDS score and stress level (R = 0.71; t(*n* − 2) = 8.40, *p* < 0.00). The scores for individual stress components between the D group and the ND group were also tested with the Mann–Whitney U test (Table 4).

### 3.4. Motivations and Problems Related to Maintaining Good Health and Physical in the D and ND Groups

Responses from the D and ND groups regarding the participants’ main motivations for undertaking regular physical activity and the problems that, in their case, represent a major obstacle to taking care of their health were explored. Interestingly, there are noticeable differences in both the motivation and problems categories. The willingness to spend time with other people significantly differentiated both groups in the motivation category (χ^2^ = 4.148, *p* = 0.042), as women in the D group were significantly less frequently motivated by this factor (21.4% vs. 45.2%). The only factor significantly differentiating the groups in the category of obstacles was lack of time (χ^2^ = 8.777, *p* = 0.003), as in the ND group, this factor was mentioned by only 14.3% of the women. In two categories, namely the receipt of medical advice and the problem of poor health, it was not possible to analyse the results with the chi-squared test due to the requirement of a minimum of five observations per subgroup not being met. Therefore, a two-sided Fisher’s test was used in these two cases. A full summary of the data analysed with the chi-squared test results is presented in Table 5.

## 4. Discussion

This study explored factors underlying the participants’ motivation to engage in regular physical activity leading to improved health and overall biopsychosocial status. Over 85% of the D group and 73% of the ND group identified the will to improve and/or maintain their physical condition as their most important motivation. Improved health and well-being were also relevant factors influencing pro-health decisions in the study group (approximately 50% in the ND group and 60% in the D group), which is in line with reports from other studies on motivations and barriers preventing older people from undertaking physical activity [51,52,53]. Interestingly, holistic health improvement and meeting friends are valued among older people, regardless of their ethnic or racial groups [54].

Lack of time, as well as lack of motivation, were the most common obstacles to undertaking health-promoting activities among the women with depressive symptoms (46.4% each). The role of physical activity in improving mental health and preventing depression is well-established in science [55,56,57,58,59]. The combination of appropriate physical activity and psychological support, information and psychoeducation is extremely important in complex problems such as obesity, high MetS risk and depressive symptoms, as confirmed by previous studies [32,39,40,41].

Older adults who engage in high levels of physical activity are at a lower risk of depression. In addition, the type of activity is important, as athletic and walker types of leisure-time activity bring the greatest benefits for mental health [60]. It is worth considering how to increase motivation in older women. Is the provision of biopsychosocial support sufficient? There is certainly also a need to develop systemic solutions to facilitate the women’s access to non-pharmacological forms of therapy and informational support. Additional research into this aspect is warranted.

Over 82% of the women in the study group were overweight or obese. Moreover, the women in the study group had a number of MetS-related health problems, such as hypertension, elevated triglycerides, reduced HDL levels, prediabetes status and diabetes, high total cholesterol and low back and joint pain. Being overweight or obese is linked to the occurrence of MetS and to adipose tissue dysfunction [61]. People with obesity are at increased risk of negative long-term outcomes even in the absence of metabolic abnormalities [62]. Certain research suggests that depression is caused by MetS and obesity, and conversely, MetS and obesity cause depression [63,64,65,66,67].

At the same time, the D group had a higher number of risk factors of MetS, which indicates the vulnerability of those patients. Similar conclusions have been drawn by the authors of other studies, in which obesity and MetS were found to be associated (independently) with depressive symptoms, irrespective of demographic factors, pharmacotherapy, behavioural factors and comorbidities [68,69,70]. However, there are also studies that have not shown an association between MetS and its components with depressive symptoms in older patients [71]. Rochlani et al. [72] stated that lifestyle modification is essential in the management of underlying risk factors. Weight reduction and subsequent maintenance of ideal body weight are crucial preventive and management strategies. It is undeniable that the incidence of MetS is closely related to obesity, ageing and unhealthy lifestyles [73,74,75].

There were significant differences between the D and ND groups in motivation and obstacles to undertaking health-promoting activities. First, the willingness to spend time with other people significantly differentiated the two groups (χ^2^ = 4.148, *p* = 0.042). Women struggling with depressive symptoms were significantly less frequently motivated by this factor. However, in the category of obstacles, the only factor significantly differentiating the two groups was lack of time (χ^2^ = 8.777, *p* = 0.003). Lack of time emerged as the second most important obstacle to physical activity in 16% of women aged 65–69 years in the Australian population study. This rate dropped to 7% in women over 70 [76].

In the ND group, lack of time was mentioned by only 14.3% of the respondents, whereas in the D group, it was mentioned by more than 46%. Several reports show lack of motivation as the most prominent barrier among middle-aged adults [77]. Furthermore, for individuals with severe mental illness, stress/depression, fatigue and insufficient support are significant barriers to undertaking physical activity [53]. Patients with psychiatric conditions report less motivation and perceive more barriers (for example, feeling more ‘tired’) than controls [78]. On the other hand, women tend to report more barriers due to caring responsibilities and housekeeping, which take precedence over physical activity [79].

Personality characteristics and family situations should also be considered, as well as the possibility that these women are so burdened by daily responsibilities that lack of time becomes a real problem in maintaining health [52]. Previous studies have not found a clear link between personality and the occurrence and development of MetS [80,81]. However, it has been noted that risk factors that form clusters may be important—for example, a combination of high levels of neuroticism, cynicism and a type D personality, which is often associated with an overall unhealthy diet or lower activity levels [81]. Unfortunately, the motivations and barriers directly related to physical activity and health-promoting activities undertaken by women with MetS and depressive symptoms are barely understood.

Nevertheless, a review focusing on motivations and barriers to physical activity in older people shows a broad characteristic of the problem. In fact, for the 65–70-year-old group (both men and women), the motivation to engage in physical activity was based on various forms of reinforcement such as encouragement from peers, having fun and receiving support from health-promotion professionals [82]. In this age group, the researchers also identified barriers of an environmental and social nature, such as lack of belief in one’s abilities, lack of family support, and family roles held by the respondents [52,82]. In terms of barriers, individual studies also point to factors such as cost, lack of facilitation and weather [82,83,84]. However, a deeper understanding of this problem in the context of the coexistence of MetS and depressive symptoms—including its causes and consequences—in further research should contribute to optimising the rehabilitation services offered to seniors at high risk of MetS and depressive symptoms.

A large-scale longitudinal study suggested that cultural resources—an individual’s knowledge, attitudes, values and behavioural norms, beliefs and skills—have a significant influence on the uneven distribution of MetS remission [85]. For example, people with higher education were more likely to achieve MetS remission, mainly because such people were more likely to exhibit healthy behaviours. On the other hand, there are indications of the role of irrational beliefs and fear in the development of MetS and the consequent need for holistic prevention focused on both mental health and MetS [86].

Older women at high risk of MetS with depressive symptoms experience higher levels of stress, both overall and in individual components (emotional tension, external stress and internal stress). They also manifest greater concerns about their health (75%) relative to women without depressive symptoms (60%). The correlation analysis did not show a strong relationship between the studied traits and stress levels and the presence and level of depressive symptoms. Only stress levels correlated strongly with the GDS scale scores, as expected—with increasing stress levels, the prevalence of depressive symptoms also increased.

Another problem is also worth noting, which is that >80% of participants in both groups were dissatisfied with the free rehabilitation available under the National Health Fund. A systemic, practical and low-cost approach to treating MetS is needed, but there still seems to be an absence of systemic solutions focused on the needs of older women with metabolic conditions and depressive symptoms [87]. Results of this study indicate the validity of conducting a multimodal therapeutic approach in rehabilitation combining physical activity with health-related psychoeducation, which is consistent with other studies focusing on lifestyle changes in the treatment of MetS in women [88,89,90,91,92].

This study also has relevant limitations. The first is the cross-sectional nature of the study, which prevents the ability to assess the existence of a causal relationship and carries a risk of bias. The second limitation is the small sample size. In addition, the motivations and obstacles identified by women with MetS and with or without depression were examined using a self-administered questionnaire, the psychometric properties of which are unknown. Our results point to a new direction for consideration and need to be confirmed by additional studies that provide strong scientific evidence.

## 5. Conclusions

Improving and/or maintaining physical fitness was identified as the greatest motivation to undertaking physical activity and health-promoting behaviour in both groups of older women with high risk of MetS and with and without depressive symptoms. Women with depressive symptoms are in great need of biopsychosocial interventions. Hence, it is important to encourage primary care physicians to screen for depression in late life. A deeper understanding of the needs of these patients is crucial, as this may ultimately lead to the development of more optimal and accessible therapeutic interventions.

## Figures and Tables

**Figure 1 ijerph-19-15957-f001:**
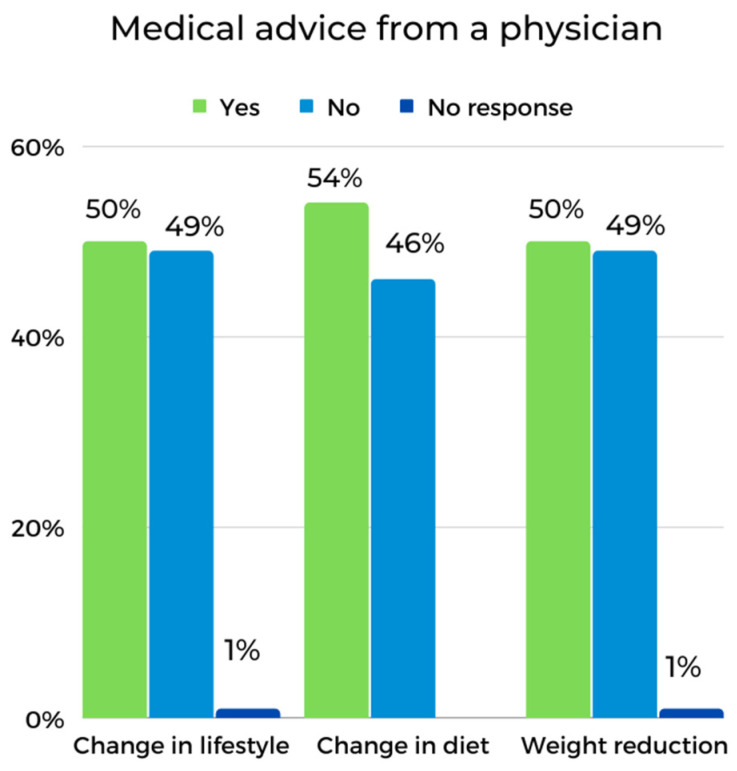
Health-related lifestyle recommendations received from a primary care physician in the study group.

**Figure 2 ijerph-19-15957-f002:**
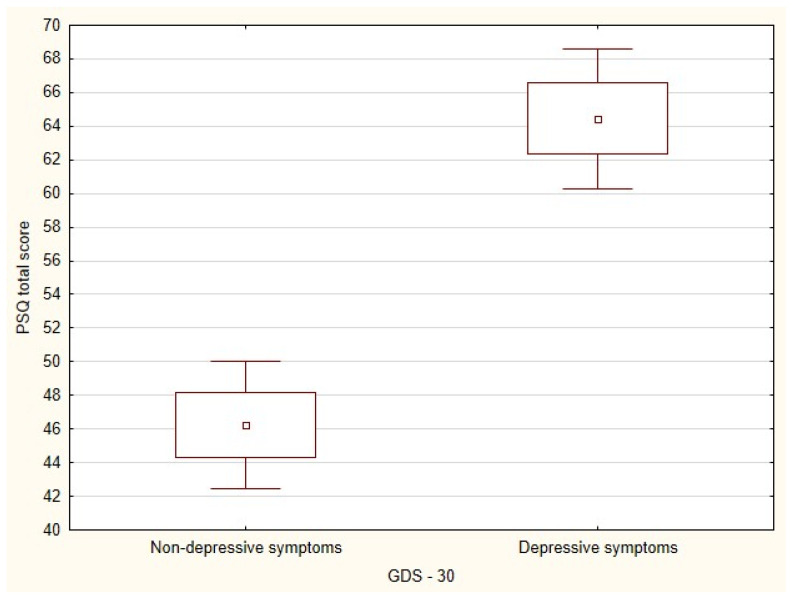
Distribution of the Perceived Stress Questionnaire scale values in relation to depressive symptom comorbidity in older women at high risk of metabolic syndrome.

**Figure 3 ijerph-19-15957-f003:**
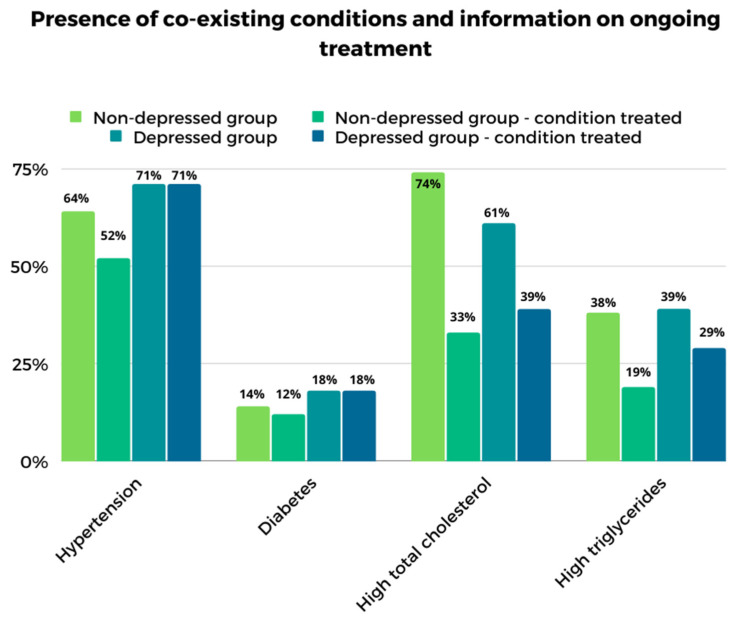
Presence of co-existing conditions and information on ongoing treatment in a study group of elderly women at high risk of metabolic syndrome in groups with and without depressive symptoms.

**Table 1 ijerph-19-15957-t001:** Baseline characteristics of the study groups of older women at high risk of metabolic syndrome with and without depressive symptoms.

	ND (*n* = 42)		D (*n* = 28)	
	Mean ± SD	Range	Mean ± SD	Range
Age (years)	70.1 ± 4.73	62–83	71.0 ± 5.66	63–84
Weight (kg)	75.1 ± 13.16	43.9–102.0	75.6 ± 19.96	44.4–135.9
Height (m)	1.59 ± 0.06	1.47–1.71	1.61 ± 0.07	1.47–1.74
BMI (kg/m^2^)	29.65 ± 4.89	19.25–39.94	29.03 ± 5.39	18.0–47.58
Waist ratio (cm)	95.9 ± 9.13	78–130	94.1 ± 12.64	61.0–120.0
Body weight classification	*n*	%	*n*	%
Normal weight	7	17	5	18
Overweight	19	45	13	46.3
Class I obesity	9	21	8	28.5
Class II obesity	7	17	1	3.6
Class III obesity	-	-	1	3.6
Education				
Basic/vocational	5	12	4	14
Secondary	21	50	16	57
Higher education	16	38	8	29
Marital status of the subject				
Married	19	45	11	39
Single	6	14	5	18
Divorced	3	7	2	7
Widow	14	34	10	36

Abbreviations: BMI—body mass index; D—depressed group; ND—non-depressed group; SD—standard deviation.

**Table 2 ijerph-19-15957-t002:** Selected clinical parameters in the study groups of women at high risk of metabolic syndrome with and without depressive symptoms.

	ND (*n* = 42)		D (*n* = 28)	
	Mean ± SD	Range	Mean ± SD	Range
Blood pressure [mmHg]				
Systolic	134.17 ± 19.73	90–186	135.4 ± 21.38	110–198
Diastolic	76.98 ± 10.51	56–114	74.1 ± 8.62	59–88
Cholesterol [mg/dL]				
Total	221.16 ± 45.96	138–342	192.48 ± 35.88	132–265
HDL	71.86 ± 13.16	44.0–183.0	63.56 ± 22.40	42.0–154.0
LDL	123.5 ± 40.50	70.0–238.0	111.3 ± 36.86	42.0–203.0
Triglycerides [mg/dL]	128.3 ± 48.56	52.0–293.0	124.2 ± 52.18	48.0–253.0
Blood sugar level [mg%]	102.5 ± 18.96	60.0–167.0	101.0 ± 14.70	78.0–140.0

Abbreviations: D—depressed group; HDL—high-density lipoprotein; LDL—low-density lipoprotein ND—non-depressed group; SD—standard deviation.

**Table 3 ijerph-19-15957-t003:** Age, education level and marital status in the study groups of women at high risk of metabolic syndrome with and without depressive symptoms.

Feature		ND Group, *n* = 42 (%)	D Group, *n* = 28 (%)	U	Z	*p*
Age		70.1 ± 4.73	71.0 ± 5.66	518.00	−0.99	0.32
Education	Basic/vocational	14.29%	11.63%	558.00	0.57	0.60
	Secondary	57.14%	53.49%			
	Higher education	28.57%	34.88%			
Marital status	Married	42.86%	46.51%	574.00	−0.32	0.74
	Single	14.29%	16.28%			
	Divorced	35.71%	32.56%			
	Widowed	35.71%	32.56%			

The groups were compared using the chi-square test (*p* < 0.05 was considered significant). Abbreviations: D—depressed group; ND—non-depressed group; U—Mann–Whitney U-test value (for *n* ≤ 20); Z—normalised U-test value (for *n* > 20).

**Table 4 ijerph-19-15957-t004:** Comparison of individual stress components between the depressed and the non-depressed group using the Mann–Whitney U test.

Stress Components	Rank Sum ND Group	Rank Sum D Group	U	Z	*p*
Emotional tension	1142.0	1414.0	196.0	−4.77	>0.0001 *
External stress	1259.5	1296.5	313.5	−3.38	0.0007 *
Intrapsychic stress	1252.0	1304.0	306.0	−3.48	0.0005 *
General PSQ score	1165.5	1390.5	219.5	−4.49	>0.0001 *

Abbreviations: D—depressed; ND—non-depressed; PSQ, Perceived Stress Questionnaire; U—Mann Whitney U-test value (for *n* ≤ 20); Z—normalised U-test value (for *n* > 20). * Statistically significant results; significance set at *p* < 0.05.

**Table 5 ijerph-19-15957-t005:** Motivations and obstacles to maintaining regular physical activity and taking care of health in older women with high metabolic syndrome risk in groups with or without depressive symptoms.

Type	ND Group	D Group	Pearson’s χ^2^	df	*p*
Improving/maintaining physical condition	73.8%	85.7%	1.414	1	0.234
Improving health	52.3%	60.7%	0.473	1	0.491
Improving well-being	50%	60.7%	0.777	1	0.378
Improving appearance/preventing obesity	45%	50%	0.153	1	0.696
Willingness to spend time with other people	45%	21%	4.148	1	0.042 *
Receiving medical advice (primary care physician)	11.9%	7.1%	Two-sided Fisher’s test		0.694
Lack of motivation	38.1%	46.4%	0.481	1	0.488
Lack of time	14.3%	46.4%	8.777	1	0.003 *
Lack of opportunity to participate	21.4%	21.4%	0.000	1	0.100
Lack of money	11.9%	25%	2.028	1	0.154
Poor health	16.7%	7.1%	Two-sided Fisher’s test		0.299

Abbreviations: D—depressed; df—degrees of freedom; ND—non-depressed. * Statistically significant results; significance set at *p* < 0.05.

## Data Availability

The data presented in this study are available upon request from the corresponding author. The data are not publicly available due to privacy restrictions.

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
