# Peer review of "Factors Associated with Undertaking Health-Promoting Activities by Older Women at High Risk of Metabolic Syndrome"

_ijerph, 2022, doi:10.3390/ijerph192315957_

Round 1

Reviewer 1 Report

Dear Authors,

Congratulations for your work.

You tried to contribute to the knowledge about health-promoting activities for elderly women at high risk of MetS. Your manuscript needs substantial improvements in discussion and results.

Here are my considerations:

Abstract:

Add introductory sentence about manuscript not only aim of the study in background

Add data of your sample in methods

Add more data in your results

Materials and methods

Line 96-99 – this information must be on sample. Rewrite this paragraph

Results and Discussion

Figure 1 and 3. Must be improved the design. Use Prism for e.g.

 Rewrite and reduce your results section.

The same for discussion. Discuss your most significant  results. (E.G. Line 450-463)

Your discussion must be centered in discuss your results with literature and not with your personal opinion.

E.g. : Line 464-479 – You must add references to support your discussion. The same to Line-480-495

Conclusion

Line 500-507 – remove

The conclusion should be reduced to just one paragraph that presents the major conclusion of your study.

Author Response

SUMMARY OF OUR RESPONSES

Thank you for your careful review of our paper as well as your comments, corrections and suggestions that ensued. A careful revision of the paper has been carried out to take all of them into account, and in the process, we believe the paper has been significantly improved. In the present „Response Letter“ we first detail the major changes that have been made in the paper to correct the main weaknesses identified by the Reviewer. We then sequentially address all the points that we have corrected „step-by-step“. Changes that have been made to the text have been marked in red. Deleted content has been crossed out.

The main changes:

  • The term “depression” has been replaced by “depressive symptoms” throughout the article,
  • The introduction section has been expanded and new references have been added,
  • The results have been reduced to the most important information,
  • The discussion has been rewritten taking into account changes that have occurred in the results and focusing on the most important reports. Also, the references have been supplemented with the most recent reports,
  • In the discussion section, the layout of the paragraphs has been also changed,
  • The summary section has been shortened. It starts with the most important results,
  • A paragraph on the limitations of the study has been added,
  • The appearance of all figures has been improved.

REVIEWER 1 EVALUATION

Dear Authors,

Congratulations for your work.

You tried to contribute to the knowledge about health-promoting activities for elderly women at high risk of MetS. Your manuscript needs substantial improvements in discussion and results.

Thank you for all your important comments. Our answers to your points are as follows:

Here are my considerations:

Abstract:

Add introductory sentence about manuscript not only aim of the study in background

Add data of your sample in methods

Add more data in your results

Response: Thank you for your important suggestion. Our abstract was originally shortened due to the 200 word limit. It has now been revised according to the reviewer's comments.

Materials and methods

Line 96-99 – this information must be on sample. Rewrite this paragraph

Response: Thank you for this comment. The paragraph has been rewritten to be based on a specific example.

Results and Discussion

Figure 1 and 3. Must be improved the design. Use Prism for e.g.

Response: The figures have been corrected in the Canva graphics programme.

Rewrite and reduce your results section. The same for discussion. Discuss your most significant  results. (E.G. Line 450-463)

Your discussion must be centered in discuss your results with literature and not with your personal opinion.

E.g. : Line 464-479 – You must add references to support your discussion. The same to Line-480-495

Response: We agree with the reviewer's comments. The results have been reduced and the discussion has been rewritten and supplemented with new references. Line 464-479  excerpt has been removed.

Conclusion

Line 500-507 – remove

The conclusion should be reduced to just one paragraph that presents the major conclusion of your study.

Response: Thank you for this meaningful comment. The excerpt has been removed, making our summary more readable.

Reviewer 2 Report

Dear authors,

In my opinion:

-In the title it is not correct that a word appears in an abbreviated form.

-The format used in the Abstract is not adequate because it uses numbering prior to each section, which must be removed.

-In keywords, “metabolic syndrome” and “depression” are missing.

-The Introduction is correctly prepared and adequately explains the research problem, with the use of updated citations. The rest of the manuscript is perfectly written and elaborated in all its sections.

As for the References section there are many errors, one is that the year of publication does not appear in bold. Reference number one is from a book and is poorly displayed, also on the web page, it does not specify the date of access to said page. In reference number 10 the DOI of the publication does not appear. In reference number 14 it repeats the year of publication. Reference number 27, 30, 34, 40 and 41 are also faulty. In general, all should be reviewed.

Author Response

SUMMARY OF OUR RESPONSES

Thank you for your careful review of our paper as well as your comments, corrections and suggestions that ensued. A careful revision of the paper has been carried out to take all of them into account, and in the process, we believe the paper has been significantly improved. In the present „Response Letter“ we first detail the major changes that have been made in the paper to correct the main weaknesses identified by the Reviewer. We then sequentially address all the points that we have corrected „step-by-step“. Changes that have been made to the text have been marked in red. Deleted content has been crossed out.

The main changes:

  • The term “depression” has been replaced by “depressive symptoms” throughout the article,
  • The introduction section has been expanded and new references have been added,
  • The results have been reduced to the most important information,
  • The discussion has been rewritten taking into account changes that have occurred in the results and focusing on the most important reports. Also, the references have been supplemented with the most recent reports,
  • In the discussion section, the layout of the paragraphs has been also changed,
  • The summary section has been shortened. It starts with the most important results,
  • A paragraph on the limitations of the study has been added,
  • The appearance of all figures has been improved.

REVIEWER 2 EVALUATION

Thank you for all your important comments. Our answers to your points are as follows:

Dear authors,

In my opinion:

-In the title it is not correct that a word appears in an abbreviated form.

Response: The abbreviation has been replaced by the full name in the title.

-The format used in the Abstract is not adequate because it uses numbering prior to each section, which must be removed.

Response: The numbering has been removed.

-In keywords, “metabolic syndrome” and “depression” are missing.

Response: Key words have been completed with missing terms.

-The Introduction is correctly prepared and adequately explains the research problem, with the use of updated citations. The rest of the manuscript is perfectly written and elaborated in all its sections.

Response: Thank you for your appreciation of our work and your comments, which help us to improve the manuscript.

As for the References section there are many errors, one is that the year of publication does not appear in bold. Reference number one is from a book and is poorly displayed, also on the web page, it does not specify the date of access to said page. In reference number 10 the DOI of the publication does not appear. In reference number 14 it repeats the year of publication. Reference number 27, 30, 34, 40 and 41 are also faulty. In general, all should be reviewed.

Response: Thank you for your valid comment. We used Zotero software when preparing our work for references, which apparently made our vigilance dormant. In the case of the cited item at No. 10 - the doi number was not assigned to the paper. Only the PMID number is available in the PubMed search. For item no. 14, this is just a coincidence as the publication was published in 2010 and the volume no. is just 2010. We made sure to bold the year of publication where we cited articles. Thanks to the reviewer's suggestion, the entire section has been carefully checked and corrected.

Reviewer 3 Report

The article aims to compare two groups of older women, with and without depression, at risk for Mets in relation to different health promoting activites, especially engagement in physical activity. Both samples are very small, far from allowing any coherent conclusion. The inclusion in the depression sample is done just on the basis of a self completed questionnaire, which I find quite incorrect, since someone cannot be declared as being depressed without a medical diagnostic. We do not really understand if the groups are just at risk of Mets, or have it already diagnosed. Also, we do not see the use of the PSQ questionnaire. Authors consider that lacking time to participate to activites is just an excuse, ignoring that family situation or SES might be a factor here. Also, the unwillingness to spend time with other people is considered as a trait for depression, but it can be connected maybe with being an extrovert or an introvert, personal characteristics that might introduce a big bias. Even more, just taking into consideration that the participation in the study was voluntary ruins any assumptions regarding the attitude of depressive patients. They might unwilling to get involved (in theory), but by volunteering to participate to the present study they contradict this assumption. For the rest, statistics is correct, references, up to date and also part of the conclusions are correct, more has to be done at grassroot level for older patients with Mets. 

Please, rewrite the article involving a trained psychologist. 

Author Response

SUMMARY OF OUR RESPONSES

Thank you for your careful review of our paper as well as your comments, corrections and suggestions that ensued. A careful revision of the paper has been carried out to take all of them into account, and in the process, we believe the paper has been significantly improved. In the present „Response Letter“ we first detail the major changes that have been made in the paper to correct the main weaknesses identified by the Reviewer. We then sequentially address all the points that we have corrected „step-by-step“. Changes that have been made to the text have been marked in red. Deleted content has been crossed out.

The main changes:

  • The term “depression” has been replaced by “depressive symptoms” throughout the article,
  • The introduction section has been expanded and new references have been added,
  • The results have been reduced to the most important information,
  • The discussion has been rewritten taking into account changes that have occurred in the results and focusing on the most important reports. Also, the references have been supplemented with the most recent reports,
  • In the discussion section, the layout of the paragraphs has been also changed,
  • The summary section has been shortened. It starts with the most important results,
  • A paragraph on the limitations of the study has been added,
  • The appearance of all figures has been improved.

REVIEWER 3 EVALUATION

Thank you for all your important comments. Our answers to your points are as follows:

The article aims to compare two groups of older women, with and without depression, at risk for Mets in relation to different health promoting activites, especially engagement in physical activity. Both samples are very small, far from allowing any coherent conclusion.

Response: We agree with the reviewer that our sample size was small, however, we have added a paragraph in the discussion section regarding the limitations of our survey and included information regarding the type of study, the small sample size and the use of an additional tool whose parameterisation indices are unknown. The questionnaire itself was translated into English and included in the supplementary materials [S1]. Nevertheless, statistically significant differences showed up between the groups, which should be further investigated.

The inclusion in the depression sample is done just on the basis of a self completed questionnaire, which I find quite incorrect, since someone cannot be declared as being depressed without a medical diagnostic. We do not really understand if the groups are just at risk of Mets, or have it already diagnosed. Also, we do not see the use of the PSQ questionnaire.

Response: Inclusion in the programme followed referral of the participant by the primary physician to the Mental Health Promotion Programme. The GDS and PSQ tests were performed and analysed by an experienced psychologist. Stress is also a risk factor in the development of MetS, and in our previous studies it also correlated strongly with the severity of depressive symptoms on the GDS scale. Some participants have a diagnosis of MetS (meeting 3 out of 5 criteria) and others are at high risk of developing MetS (presence of 2 out of 5 factors). We obtained these details from the primary physician. We have completed this information in the Materials and Methods section. A distribution of MetS components present in both groups was included in Table 6, but in line with the reviewers' comment to reduce the section to include only the most important and relevant results - the table has been removed in the revised version. We hope that the manuscript is improved in the current version.

Table 6. The distribution of MetS components achieved by the subjects according to the co-occurrence of depression.

Number of MetS components

Normal Mood

(n = 42)

%

Depression

(n = 28)

%

2

18

43

10

36

3

14

33

9

32

4

8

19

6

21

5

2

5

3

11

Abbreviations: MetS – metabolic syndrome; Components: triglyceride levels, fasting sugar, waist circumference in cm, blood pressure and HDL – high-density lipoprotein levels.

Authors consider that lacking time to participate to activites is just an excuse, ignoring that family situation or SES might be a factor here. Also, the unwillingness to spend time with other people is considered as a trait for depression, but it can be connected maybe with being an extrovert or an introvert, personal characteristics that might introduce a big bias. Even more, just taking into consideration that the participation in the study was voluntary ruins any assumptions regarding the attitude of depressive patients. They might unwilling to get involved (in theory), but by volunteering to participate to the present study they contradict this assumption. For the rest, statistics is correct, references, up to date and also part of the conclusions are correct, more has to be done at grassroot level for older patients with Mets. 

Please, rewrite the article involving a trained psychologist. 

Response: We agree with the reviewer that we made a mistake here. Thank you for this feedback. As a result, we have significantly revised the discussion sections - we have removed the less relevant threads, added the most recent testimonials and introduced the comments highlighted by the reviewer. We hope that with these corrections, our work has improved significantly and will be a first step in exploring the topic of motivation and obstacles in women who have already decided to make lifestyle changes. Because of the practical dimension of the Mental Health Promotion Programme, this is a key aspect for us, helping us to better understand the women who come to us for help, which at the same time results in optimising the therapeutic programme offered. Other factors analysed in this study are also related to the practical dimension of the Mental Health Promotion Programme - perceived pain, satisfaction with physical condition and weight, advice from the primary physician. We believe that all these elements are crucial for a deep understanding of the patients we work with and the safe carrying out of therapeutic interventions. Following the reviewer's comment, we have expanded the introduction section to make it easier to read and understand further results.

Reviewer 4 Report

The authors conducted a cross-sectional observational study to examine factors associated with undertaking health-promoting activity in older women at high risk of metabolic syndrome. The authors analyzed data from 70 women who met at least two metabolic syndrome criteria and participated in an intervention program. They showed that participants with depressed moods had higher stress levels and were more likely to have a lack of time to maintain regular physical activity and take care of their health. There are some comments.

Comments:

1.      Abstract (methods): Please provide the number of participants and list the factors/characteristics measured by the self-developed questionnaire.

2.      Abstract (conclusion): I would suggest concluding with a summary of this study’s specific findings.

3.      Introduction: This study focused on women. However, the rationale was not clear. Why not men? A more explicit description of the rationale is recommended.

4.      Introduction: This study examined factors associated with undertaking health-promoting activity. There was a missing part in the introduction. According to the literature, what were the possible barriers to undertaking health-promoting activity in women at risk of metabolic syndrome? This study examined multiple factors, including depressive mood, lifestyles, stress, metabolic disorders, medical conditions, pain, recommendations from the primary care physician, etc. A rationale for examining these factors is lacking.

5.      Methods (Line 132 on Page 3): “participate in the study on a voluntary basis.” How many participants met the inclusion and exclusion criteria and were eligible to enter the study? How many participants refused to join the study? And, what were the differences in characteristics between those who entered the study (n=70) and those who did not? I suggest adding a table to present and compare the characteristics of those who entered the study and those who did not. Finally, how would the differences affect the generalizability of this study’s findings? If the data were unavailable, I would recommend discussing the issue in DISCUSSION.

6.      Methods: I would suggest replacing “2.4. Methods” with “2.4. Measurement”.

7.      Methods (Line 141-150 on Page 4): Please provide the definition (GDS score >19?) of depression used in this study.

8.      Methods (Line 182-184 on Page 5): “questions concerning the reason why the respondents decided to participate in the project (6 questions) and the obstacles that have prevented them from taking care of their health so far (5 questions).” Because these questionnaires measured key variables of this study, I suggest providing more detailed information about these questionnaires, including the questions, the items, and their psychometric properties (if available). In case their psychometric properties are unknown, a discussion of this limitation in DISCUSSION is recommended.

9.      Methods (Data analysis): Have the authors considered regression analysis adjusting for potential confounders while examining the relationship between depression and other factors/characteristics? For instance, the authors may want to determine whether depressed participants were more likely to report a “lack of time” simply because of a direct association between the depressed mood and “lack of time.” In this case, depression could be regressed on “lack of time” and confounding characteristics that may contribute to both “lack of time” and depressed mood (e.g., stress).

10.  Results (Figure 3 and Table 6): Conducting statistical test (depressed vs non-depressed) and presenting the test results are recommended.

11.  Discussion (Line 433-454 on Page 12 and Line 455-479 on Page 13): The authors discussed their findings regarding the relationships of depressed mood with motivation and obstacles to undertaking pro-health activities, stress, and treatment of health conditions. I would suggest reporting/summarizing the results from previous similar studies. Were they consistent with the findings of the current study?

12.  Discussion: A major limitation of this study is that it is cross-sectional in nature. A discussion is recommended.

13.  Conclusion (Page 13): I would suggest beginning with a summary of this study’s specific findings.

Author Response

SUMMARY OF OUR RESPONSES

Thank you for your careful review of our paper as well as your comments, corrections and suggestions that ensued. A careful revision of the paper has been carried out to take all of them into account, and in the process, we believe the paper has been significantly improved. In the present „Response Letter“ we first detail the major changes that have been made in the paper to correct the main weaknesses identified by the Reviewer. We then sequentially address all the points that we have corrected „step-by-step“. Changes that have been made to the text have been marked in red. Deleted content has been crossed out.

The main changes:

  • The term “depression” has been replaced by “depressive symptoms” throughout the article,
  • The introduction section has been expanded and new references have been added,
  • The results have been reduced to the most important information,
  • The discussion has been rewritten taking into account changes that have occurred in the results and focusing on the most important reports. Also, the references have been supplemented with the most recent reports,
  • In the discussion section, the layout of the paragraphs has been also changed,
  • The summary section has been shortened. It starts with the most important results,
  • A paragraph on the limitations of the study has been added,
  • The appearance of all figures has been improved.

REVIEWER 4 EVALUATION

Thank you for all your important comments. Our answers to your points are as follows:

The authors conducted a cross-sectional observational study to examine factors associated with undertaking health-promoting activity in older women at high risk of metabolic syndrome. The authors analyzed data from 70 women who met at least two metabolic syndrome criteria and participated in an intervention program. They showed that participants with depressed moods had higher stress levels and were more likely to have a lack of time to maintain regular physical activity and take care of their health. There are some comments.

Comments:

  1. Abstract (methods): Please provide the number of participants and list the factors/characteristics measured by the self-developed questionnaire.

Response: The data has been completed.

  1. Abstract (conclusion):I would suggest concluding with a summary of this study’s specific findings.

Response: We agree with the reviewer. We have made the proposed changes to the summary section.

  1. Introduction:This study focused on women. However, the rationale was not clear. Why not men? A more explicit description of the rationale is recommended.

Response: Our study was carried out as part of the Mental Health Promotion Programme at the Foundation for Senior Citizen Activation SIWY DYM, which focuses on biopsychosocial support for women over 60. We decided to add a paragraph in the introduction section that better explains the specifics of the study and shows that the therapeutic programme and the various activities of the foundation are aimed at elderly women. We hope that this improves the clarity of the article.

  1. Introduction:This study examined factors associated with undertaking health-promoting activity. There was a missing part in the introduction. According to the literature, what were the possible barriers to undertaking health-promoting activity in women at risk of metabolic syndrome? This study examined multiple factors, including depressive mood, lifestyles, stress, metabolic disorders, medical conditions, pain, recommendations from the primary care physician, etc. A rationale for examining these factors is lacking.

Response: Unfortunately, most of the research related to MetS and depression focuses on lifestyle understood as regular physical activity and a healthy diet or risk factors, but we did not find studies looking for motivations or obstacles related to changes in habits in women who showed a willingness to make such changes. Because of the practical dimension of the Mental Health Promotion Programme, this is a key aspect for us, helping us to better understand the women who come to us for help, which at the same time results in optimising the therapeutic programme offered. Other factors analysed in this study are also related to the practical dimension of the Mental Health Promotion Programme - perceived pain, satisfaction with physical condition and weight, advice from the primary physician. We believe that all these elements are crucial for a deep understanding of the patients we work with and the safe carrying out of therapeutic interventions. Following the reviewer's comment, we have expanded the introduction section to make it easier to read and understand further results.

  1. Methods (Line 132 on Page 3): “participate in the study on a voluntary basis.” How many participants met the inclusion and exclusion criteria and were eligible to enter the study? How many participants refused to join the study? And, what were the differences in characteristics between those who entered the study (n=70) and those who did not? I suggest adding a table to present and compare the characteristics of those who entered the study and those who did not. Finally, how would the differences affect the generalizability of this study’s findings? If the data were unavailable, I would recommend discussing the issue in DISCUSSION.

Response: Patients were recruited to the project by their primary care physicians based on the results of medical tests and anthropometric measurements. We do not have information on the number of people who had been screened but did not meet the inclusion criteria for the project. Participation in the study on a voluntary basis means that the participants had the choice to take part in the programme offered by the Foundation for Senior Citizen Activation SIWY DYM. We have added the relevant information to the section: „Inclusion Criteria for the Research Project”.

  1. Methods: I would suggest replacing “2.4. Methods” with “2.4. Measurement”.

Response: The change has been implemented.

  1. Methods (Line 141-150 on Page 4): Please provide the definition (GDS score >19?) of depression used in this study.

Response: In the measurement section, we have described that: “A score between 0–9 points indicates no depressive symptoms, a score between 10–19 points indicates mild depression, and a score greater than 19 points represents a severe form of depression.” .In our study, a participant's GDS score of more than 9 pts was associated with increasing depressive symptoms and allocation to group D (depressive symptoms group). We have added this clarification in the 'participants' section in line with the reviewers' comments. We hope that the now revised sections are no longer in doubt.

  1. Methods (Line 182-184 on Page 5):“questions concerning the reason why the respondents decided to participate in the project (6 questions) and the obstacles that have prevented them from taking care of their health so far (5 questions).” Because these questionnaires measured key variables of this study, I suggest providing more detailed information about these questionnaires, including the questions, the items, and their psychometric properties (if available). In case their psychometric properties are unknown, a discussion of this limitation in DISCUSSION is recommended.

Response: Thank you for your valuable comment. The content of the questionnaire has been translated into English and is included in the supplementary materials. Due to unknown psychometric properties (the original purpose of the questionnaire is to supplement clinical data in order to optimise the therapeutic activities of the foundation), we decided to discuss this limitation in the discussion section.

  1. Methods (Data analysis): Have the authors considered regression analysis adjusting for potential confounders while examining the relationship between depression and other factors/characteristics? For instance, the authors may want to determine whether depressed participants were more likely to report a “lack of time” simply because of a direct association between the depressed mood and “lack of time.” In this case, depression could be regressed on “lack of time” and confounding characteristics that may contribute to both “lack of time” and depressed mood (e.g., stress).

Response: In our statistical calculations, we used linear regression taking into account the scores obtained from the parametric tests. GDS significantly correlates only with lack of time, lack of time does not correlate with other predictors. Analysis of variance (linear regression) does not describe the model better than the arithmetic mean, as F(5,64)= 2.15; p <0.07020. Adjusted R2 = 0.0772; i.e. the model explains 7% of the variance. Therefore, we felt it was appropriate to dispense with additional description of this analysis, to present the results more clearly.

  1. Results (Figure 3 and Table 6): Conducting statistical test (depressed vs non-depressed) and presenting the test results are recommended.

Response: Statistical tests were performed in both cases, which showed no significant differences between the groups. However, these differences were there, we wanted to present them. Following advice from receivers to shorten the results section and present only the most important data, we decided to remove these results in the revised manuscript.

  1. Discussion (Line 433-454 on Page 12 and Line 455-479 on Page 13): The authors discussed their findings regarding the relationships of depressed mood with motivation and obstacles to undertaking pro-health activities, stress, and treatment of health conditions. I would suggest reporting/summarizing the results from previous similar studies. Were they consistent with the findings of the current study?

Response: Unfortunately, we could not compare our results directly with others due to the lack of reports. Nevertheless, the entire discussion section has been rewritten, complete with new references and new threads that have improved its coherence and value.

  1. Discussion: A major limitation of this study is that it is cross-sectional in nature. A discussion is recommended.

Response: Thank you for this comment. Definitely the introduction of the paragraph about the limitations of the study was necessary. As suggested by the reviewer, we have completed this information.

  1. Conclusion (Page 13): I would suggest beginning with a summary of this study’s specific findings.

Response: We agree with the reviewer. We have made the proposed changes to the summary section.

Reviewer 5 Report

Dear Editors and authors,

Thank you for inviting me to review this manuscript, and congratulations for your work. This is a relevant exploration to understand the factors associated with health-promoting activities in a population of elderly women at high risk of metabolic syndrome. However, there are some opportunities for improvement, as is outlined below.

TITLE – KEYWORDS

- It would be advisable to include the term “depression” or "depressive symptoms" either in the title or in the keywords.

ABSTRACT

-In the lines 23 and 24 the correct form may be “It is important to encourage” instead of “it is important to encouraging”. 

-In line 24 “old-age depression” maybe should be substituted by “late life depression”.

INTRODUCTION

-In line 50, “as” is repeated two times, it may be a writing mistake.

-In line 51, it is said “visceral obesity and carbohydrate”, it looks like a word may be missing. “Carbohydrate consumption” may be the the term you were referring to.

-In lines 68-71, it would be advisable to add a reference or various references.

-Lines 76-79 may be worth to be written differently in order to express the ideas in a clearer manner.

METHODS

-Line 125, the term “steps” could be substituted by “groups”.

-Lines 129-132 may need to get restructured in order to make the information about participants clearer.

-In lines 141-150, the term “mood” may not be the most suitable. As Geriatric Depression Scale is a screening depression tool, it could be more appropriate to use the term "depressive symptoms". Thus, in the line 144 “0–9 points indicates normal mood” could be replaced for “0–9 points indicates no depressive symptoms”, the same occurs in the line 142 where “was used to assess mood” could be replaced for “was used to assess depressive symptoms and depression”.

-Besides it being mentioned in the abstract, it would be advisable to include in the point “2.3.Participants” which are the exact criteria to include participants in the D group or in the ND group. For instances, “participants with scores over 9 points in the GDS were included in the D group”.

RESULTS

-In the line 261, as mentioned before in the methods, the term “mood” may not be the most appropriate when referring to the GDS results, as depression is a complex and a multifactorial phenom that the word “mood” may not be enough to describe it. Thus “area of mood and levels of depression” may be replaced for “depression and depressive symptoms”.

- In Figure 2, “normal mood” should be replaced for “non-depressive symptoms”.

-In “Table 5” and “Table 7”, please add the explanations for the abbreviations “U” and “Z”.

-In Table 6, “normal mood” should be replaced for “non-depressive symptoms”.

DISCUSSION

-In the line 401, “normal mood levels” may need to be replaced by “non-depressive symptoms”.

- In the line 434, “normal mood” may need to be replaced by “non-depressive symptoms”.

-In the paragraph from line 419 to 431 and the paragraph from line 433 to 448, please compare your results to other available research about motivation and obstacles to undertaking health-promoting activities among women with depression and without depression.

-In the line 461, it may be more appropriate to change “mood” for “depression” o “depressive symptoms”. The same occurs in line 462, where “mood deteriorated”, may be changed for, “depressive symptoms increased their prevalence”.

-In the lines 469 to 477, “The health problems associated with MetS and depression are compounded by high levels of stress. Change and the need to adapt to biological, physical, environmental and social demands are the main reasons for experiencing stress. The life changes experienced by older people become stressors that negatively affect their health and functioning. Although not in every case experiencing high levels of stress is associated with the occurrence of depression, such individuals are at high risk of developing depressive symptoms. Also, the postmenopausal period in women can also have a significant impact on the onset of adaptive disorders leading to high levels of stress.”, these statements need to be accompanied by the corresponding references.

Author Response

SUMMARY OF OUR RESPONSES

Thank you for your careful review of our paper as well as your comments, corrections and suggestions that ensued. A careful revision of the paper has been carried out to take all of them into account, and in the process, we believe the paper has been significantly improved. In the present „Response Letter“ we first detail the major changes that have been made in the paper to correct the main weaknesses identified by the Reviewer. We then sequentially address all the points that we have corrected „step-by-step“. Changes that have been made to the text have been marked in red. Deleted content has been crossed out.

The main changes:

  • The term “depression” has been replaced by “depressive symptoms” throughout the article,
  • The introduction section has been expanded and new references have been added,
  • The results have been reduced to the most important information,
  • The discussion has been rewritten taking into account changes that have occurred in the results and focusing on the most important reports. Also, the references have been supplemented with the most recent reports,
  • In the discussion section, the layout of the paragraphs has been also changed,
  • The summary section has been shortened. It starts with the most important results,
  • A paragraph on the limitations of the study has been added,
  • The appearance of all figures has been improved.

REVIEWER 5 EVALUATION

Dear Editors and authors,

Thank you for inviting me to review this manuscript, and congratulations for your work. This is a relevant exploration to understand the factors associated with health-promoting activities in a population of elderly women at high risk of metabolic syndrome. However, there are some opportunities for improvement, as is outlined below.

Thank you for all your important comments. Our answers to your points are as follows:

TITLE – KEYWORDS

- It would be advisable to include the term “depression” or "depressive symptoms" either in the title or in the keywords.

Response: We are grateful for this comment. We agree with the reviewer that a more appropriate expression than depression for our study would be depressive symptoms. Changes have been made throughout the paper.

ABSTRACT

-In the lines 23 and 24 the correct form may be “It is important to encourage” instead of “it is important to encouraging”. 

-In line 24 “old-age depression” maybe should be substituted by “late life depression”.

Response: Thank you for your relevant suggestions. We have followed them.

INTRODUCTION

-In line 50, “as” is repeated two times, it may be a writing mistake.

-In line 51, it is said “visceral obesity and carbohydrate”, it looks like a word may be missing. “Carbohydrate consumption” may be the the term you were referring to.

-In lines 68-71, it would be advisable to add a reference or various references.

-Lines 76-79 may be worth to be written differently in order to express the ideas in a clearer manner.

Response: The extract has been reorganised and one reference from the biopsychosocial concept has been added. We hope that we have been able to express the ideas more clearly. Thank you for highlighting your concerns about the introduction section. They have helped us to correct errors and improve the quality of the content presented.

METHODS

-Line 125, the term “steps” could be substituted by “groups”.

-Lines 129-132 may need to get restructured in order to make the information about participants clearer.

-In lines 141-150, the term “mood” may not be the most suitable. As Geriatric Depression Scale is a screening depression tool, it could be more appropriate to use the term "depressive symptoms". Thus, in the line 144 “0–9 points indicates normal mood” could be replaced for “0–9 points indicates no depressive symptoms”, the same occurs in the line 142 where “was used to assess mood” could be replaced for “was used to assess depressive symptoms and depression”.

-Besides it being mentioned in the abstract, it would be advisable to include in the point “2.3.Participants” which are the exact criteria to include participants in the D group or in the ND group. For instances, “participants with scores over 9 points in the GDS were included in the D group”.

Response: Thank you for your comment regarding the methods section. We have completed the 'Participants' section with the suggested information and we have followed all the suggested changes.

RESULTS

-In the line 261, as mentioned before in the methods, the term “mood” may not be the most appropriate when referring to the GDS results, as depression is a complex and a multifactorial phenom that the word “mood” may not be enough to describe it. Thus “area of mood and levels of depression” may be replaced for “depression and depressive symptoms”.

- In Figure 2, “normal mood” should be replaced for “non-depressive symptoms”.

-In “Table 5” and “Table 7”, please add the explanations for the abbreviations “U” and “Z”.

-In Table 6, “normal mood” should be replaced for “non-depressive symptoms”.

Response: We agree with the reviewer's opinion. Thank you for your constructive comments. Explanations for the abbreviations U and Z have been added in both tables. All suggestions have been taken into account in the new version of the paper.

DISCUSSION

-In the line 401, “normal mood levels” may need to be replaced by “non-depressive symptoms”.

- In the line 434, “normal mood” may need to be replaced by “non-depressive symptoms”.

-In the paragraph from line 419 to 431 and the paragraph from line 433 to 448, please compare your results to other available research about motivation and obstacles to undertaking health-promoting activities among women with depression and without depression.

Response: Unfortunately, we could not compare our results directly with others due to the lack of reports. Nevertheless, the entire discussion section has been rewritten, complete with new references and new threads that have improved its coherence and value.

-In the line 461, it may be more appropriate to change “mood” for “depression” o “depressive symptoms”. The same occurs in line 462, where “mood deteriorated”, may be changed for, “depressive symptoms increased their prevalence”.

-In the lines 469 to 477, “The health problems associated with MetS and depression are compounded by high levels of stress. Change and the need to adapt to biological, physical, environmental and social demands are the main reasons for experiencing stress. The life changes experienced by older people become stressors that negatively affect their health and functioning. Although not in every case experiencing high levels of stress is associated with the occurrence of depression, such individuals are at high risk of developing depressive symptoms. Also, the postmenopausal period in women can also have a significant impact on the onset of adaptive disorders leading to high levels of stress.”, these statements need to be accompanied by the corresponding references.

Response: Due to the reduction of the results section and the rewriting of the discussion focusing only on the most important results, we have decided to remove this section.

Round 2

Reviewer 1 Report

Dear authors,

you have attended some of my comsiderations. Your article improved in some parts but it still need significant improvements. Your article is written like a report.

Abstract:

Add data of your  sample in methods not in results

Line 24-25 add significance

Line 28-30 Rewrite. This is an article not a report. You cannot write "we identified....." rewrite to something like this: "Motivations and barriers to undertake health-promoting activities and levels of perceived stress were different between booth groups"

Methods:

Line 125 - add years after the data and remove "mean"

Change table 1 and add baseline values  of booth groups not all participants together.

Results:

Don´t write the article like an report.

E.g. Line 260 - ...we identified...

Discussion

Don´t write the article like an report.

e.g Line 538; 579; 587; 

Remove LINE 399-403 

Line 587-593 - Add reference to support this.

Line 598-602 You cannot state there are no other studies about this. Did you search all databases?

You must find more studies to discuss you results. This section needs literature to strenght your work.

Conclusions

Remove Line 637 - You cannot state "...us..."

Rewrite your conclusions....What you wrote are not conclusions. They are results....resume to a single paragraph....

Author Response

SUMMARY OF OUR RESPONSES

We would like to thank you for your careful review of our paper again, as well as for the corrections and suggestions that resulted from it. A careful revision of the paper has been carried out taking into account the new reviewer's comments. In this 'Response Letter', we first detail the main changes that have been made to the paper to correct the main weaknesses identified by the Reviewer. We then refer to all the points that we have improved in turn, 'step by step'. Changes that have been made to the text are marked in yellow. Deleted content has been crossed out.

The main changes:

  • The discussion has been revised according to the reviewer's advice, literature has been added and the theme of barriers and motivations to undertake physical activity occurring in older people has been expanded,
  • The summary section has been revised,
  • We have paid more attention to the writing style of the article so that it does not resemble a report,
  • Baseline characteristics of the study group in Table 1 were divided into groups with and without depressive symptoms.

REVIEWER 1 EVALUATION

Dear authors,

you have attended some of my comsiderations. Your article improved in some parts but it still need significant improvements. Your article is written like a report.

Abstract:

Add data of your  sample in methods not in results

Line 24-25 add significance

Line 28-30 Rewrite. This is an article not a report. You cannot write "we identified....." rewrite to something like this: "Motivations and barriers to undertake health-promoting activities and levels of perceived stress were different between booth groups"

Response: The passages mentioned have been corrected. In the abstract, we removed the mention of more MetS components in women with depressive symptoms, as although differences were apparent, they were not statistically significant. In an earlier version, there was a table (No. 6) in which the results were presented as a percentage, but this was removed after the first revision. Consequently, the sentence that appeared in the abstract was also removed.

Methods:

Line 125 - add years after the data and remove "mean"

Change table 1 and add baseline values  of booth groups not all participants together.

Results:

Don´t write the article like an report.

E.g. Line 260 - ...we identified...

Response: Thank you for pointing out the style of the article. We have focused more attention on it this time and hope to have eliminated the questionable sentences. Smaller suggested changes have also been made to the paper.

Discussion

Don´t write the article like an report.

e.g Line 538; 579; 587; 

Remove LINE 399-403 

Line 587-593 - Add reference to support this.

Line 598-602 You cannot state there are no other studies about this. Did you search all databases?

You must find more studies to discuss you results. This section needs literature to strenght your work.

Response: In a previous version of this article, we focused on barriers and motivations only in a group of older women with MetS. As we wrote in the introduction, the great majority of studies focus on the importance of lifestyle and diet, which is also understandable given the recommendations for MetS prevention. Thanks to the reviewer's suggestion, we decided to consider this concept far more broadly. As a result, articles on both motivations and obstacles to physical activity that occur in older people, in women and in people with psychiatric symptoms were included in the discussion. In our opinion, this definitely improved the quality of the discussion and influenced the overall perception of the work. We particularly thank you for your comments on the discussion section. 

Conclusions

Remove Line 637 - You cannot state "...us..."

Rewrite your conclusions....What you wrote are not conclusions. They are results....resume to a single paragraph....

Response: The summary has been corrected. It is limited to one paragraph and summarises the most important information. It is hoped that, with the amendments, this section is significantly improved. 

We believe that the changes made have significantly improved the quality of our work. Once again, we would like to thank you for your insightful review and suggestions that have helped us in this process.

Reviewer 3 Report

The authors have answered adequately at all our questions and is adequate for publication.

Author Response

We believe that the changes made have significantly improved the quality of our work. Once again, we would like to thank you for your insightful review and suggestions that have helped us in this process.

Reviewer 4 Report

All the concerns have been addressed. 

Author Response

(The authors gave the same response as above.)

Round 3

Reviewer 1 Report

Dear Authors,

your language in your article continues to need significant changes. You use diferent verb tenses in the same sentence. Its difficult to read in some parts. (e.g. 158-161).

1. You abstract is confusing. E.G: Its impossible to understand where are the groups of your study. you have two groups and in your methods there is only one.

2. Some  results are in methods and should be in the results (e.g. Hypertension and triglycerides values)

3. Your discussion is hard to read. 

4. Conclusion continues to be to long and not resuming your results.

Author Response

SUMMARY OF OUR RESPONSES

We would like to thank you for your careful review of our paper again. A careful revision of the paper has been carried out taking into account the reviewer's new comments. In this 'Response Letter' we refer to all the points that we have improved in turn, 'step by step'. Changes that have been made to the text are marked in yellow. Deleted content has been crossed out.

REVIEWER 1 EVALUATION

Dear Authors,

your language in your article continues to need significant changes. You use diferent verb tenses in the same sentence. Its difficult to read in some parts. (e.g. 158-161).

Response: Due to the reviewer's comments regarding significant language corrections, the paper has undergone professional ProofReading (Cambridge University ProofReading) and the certificate will be available on Monday.

  1. You abstract is confusing. E.G: Its impossible to understand where are the groups of your study. you have two groups and in your methods there is only one.

Response: In our abstract, information about the entire study group appears first, as the division was only applied after the psychologist had analysed the results of the GDS scale. However, we removed the sentence about obesity, which applied to the whole group and could have been confusing for the reader.

  1. Some results are in methods and should be in the results (e.g. Hypertension and triglycerides values)

Response: In line with the reviewer's comment, we have removed some of the data (hypertension, Blood sugar level and triglycerides values from Table 1 and moved it to a new Table 2 located at the beginning of the results section.

  1. Your discussion is hard to read.

Response: The entire article has been carefully checked and linguistically corrected.  Due to the reviewer's comments regarding significant language corrections, the paper has undergone professional ProofReading (Cambridge University ProofReading) and the certificate will be available on Monday. We hope that this has significantly improved the perception of our paper.

  1. Conclusion continues to be to long and not resuming your results.

Response: The summary section has been modified.  It includes four sentences and summarises the important findings for this article and the proposed future research directions.